# The Agreement of a Two- and a Three-Dimensional Speckle-Tracking Global Longitudinal Strain

**DOI:** 10.3390/jcm11092402

**Published:** 2022-04-25

**Authors:** Jiří Plášek, Tomáš Rychlý, Diana Drieniková, Ondřej Cisovský, Tomáš Grézl, Miroslav Homza, Jan Václavík

**Affiliations:** 1Department of Internal Medicine and Cardiology, University Hospital Ostrava, 708 52 Ostrava, Czech Republic; diana.drienikova@fno.cz (D.D.); tomasgrezl@gmail.com (T.G.); jan.vacalvik@fno.cz (J.V.); 2Benedor Cardiology Outpatient Clinic Ltd., 708 00 Ostrava, Czech Republic; info@benedor.cz (T.R.); ondrej.ciso@gmail.com (O.C.); mirek.homza@centrum.cz (M.H.); 3Faculty of Medicine, University of Ostrava, 703 00 Ostrava, Czech Republic

**Keywords:** global longitudinal strain, 17-segment AHA model, deformation imaging, three-dimensional echocardiography, speckle-tracking echocardiography

## Abstract

Background: Two-dimensional (2D) and three-dimensional (3D) speckle-tracking echocardiography (STE) enables assessment of myocardial function. Here, we examined the agreement between 2D and 3D STE measurement of a global longitudinal strain (GLS) in patients with normal left ventricle, reduced ejection fraction, and cardiac pacing. Methods: Our analysis included 90 consecutive patients (59% males; average age: 73.2 ± 11.2 years) examined between May 2019–December 2020, with valid 2D and 3D loops for further speckle-tracking strain analysis. Linear regression, Pearson correlation, and a Bland–Altman plot were used to quantify the association between 2D and 3D GLS and related segments, using the 17-segment American Heart Association (AHA) model. Analyses were performed in the entire study group and subgroups. Intra- and inter-observer variability of 2D and 3D GLS measurement was also performed in all participants. Results: We observed a strong correlation between 2D and 3D GLS measurements (R = 0.76, *p* < 0.001), which was higher in males (R = 0.78, *p* < 0.001) than females (R = 0.69, *p* < 0.001). Associated segment correlation was poor (R = 0.2–0.5, *p* < 0.01). The correlation between 2D and 3D GLS was weaker in individuals with ventricular pacing of >50% (R = 0.62, *p* < 0.001) than <50% (R = 0.8, *p* < 0.001), and in patients with LVEF of <35% (R = 0.69, *p* = 0.002) than >35% (R = 0.72, *p* < 0.001). Intra-observer variability for 2D and 3D GLS was 2 and 2.3%, respectively. Inter-observer variability for 2D and 3D GLS was 3.8 and 3.6%, respectively Conclusion: Overall 2D and 3D GLS were closely associated but not when analyzed per segment. It seems that GLS comparison is more representative of global shortening than local displacement. Right ventricular pacing and reduced left ventricular ejection fraction were associated with a reduced correlation between 2D and 3D GLS.

## 1. Introduction

Speckle-tracking echocardiography (STE) is a promising method for non-invasive myocardial deformation analysis [1]. Compared to magnetic resonance imaging (MRI), two-dimensional (2D) STE enables angle-independent and reliable measurement of left ventricular dimensions and strains [1]. Despite years of research showing advantages over conventional parameters, 2D STE is not commonly used in clinical practice, except for cardio-oncology [2]. Reasons include that the analysis is time-consuming, a lack of standardization, inter-vendor differences, and the need for manual adjustments to the cardiac regions of interest [3,4,5]. Moreover, different modalities such as 2D STE, three-dimensional (3D) STE, and cardiac MRI are available to acquire myocardial strain measurements, raising questions regarding the agreement between methods [4,5].

Two-dimensional STE has been validated against MRI tagging, as a gold standard of deformation analysis [1,6]. However, based on the expert consensus statement, there is no true gold standard technique for non-invasive quantification of left ventricular (LV) mechanics [7]. MRI tagged myocardial strain showed an excellent correlation to 2D STE [1], similar to the agreement previously described between tissue Doppler imaging (TDI)-based strain and MRI-tagging [8].

Two-dimensional STE enables feasible assessment of global and regional myocardial function [5]. The model is reconstructed and segmented from three 2D planes, in contrast to 3D volumetric speckle-tracking analysis. Three-dimensional STE is an emerging ultrasonographic modality that may provide us with more physiological, and probably faster, analysis of myocardial deformation. The results of 3D STE should be cautiously evaluated. Compared to 2D STE, 3D STE involves a considerably lower average frame rate and a higher level of automatization of the analysis. Therefore, there remains a need to examine the agreement between 2D and 3D STE.

Global longitudinal strain (GLS) predominantly reflects the contractile function of the subendocardium and subepicardium of the left ventricular wall due to myofiber orientation [9]. Notably, the subendocardium is more susceptible to both ischemia/stunning and mechanical overload related to either valvular disease or aging [9,10,11]. Therefore, GLS is likely to decrease in early stages of various cardiac diseases [9]. GLS is rarely systematically used, although it is reliable, and it has a reproducible parameter, even when compared to left ventricular ejection fraction (LVEF) [12].

Based on its reliability, sensitivity, and reproducibility, GLS was the main parameter investigated in our present study. We aimed to analyze the level of agreement between GLS measured by 2D vs. 3D STE and its reproducibility. Gender-related influences on left ventricular mechanics in individuals free of heart failure have been recently described [13]. Therefore, separate analyses for males and females were also performed.

## 2. Materials and Methods

### 2.1. Patients

For this study, we retrospectively enrolled echocardiographic examinations from 90 consecutive patients, who were scheduled for routine evaluation. Incomplete loops and/or inadequate image quality for 2D and 3D STE led to exclusion from the study. Patients with atrial fibrillation at the time of the image acquisition were also excluded from the study. The study sample comprised of patients with various cardiovascular diseases to assess the correlation between 2D and 3D STE across a real-life patient population. This study was approved by the institutional review board of University Hospital Ostrava and conducted in accordance with the Helsinki Declaration. The need for informed consent was waived for this study.

### 2.2. Echocardiography

Two-dimensional grayscale echocardiography was performed using a Vivid E95 scanner (GE Vingmed Ultrasound, Horten, Norway). The frame rate was >50/s for 2D STE, and >25/s for 3D STE. Images were analyzed using EchoPAC version 203 revision 73 (GE Vingmed Ultrasound, Horten, Norway). The endocardial border was traced at end-systole, and the thickness of the region of interest (ROI) was adjusted to include most of the myocardium, while avoiding stationary speckles near the pericardium. From the 3060 analyzed segments of the 17-segment AHA model, a total of 136 segments (4.4%) were excluded from the analysis due to an inability to track.

For 2D STE, we used automatic function imaging (AFI) of the EchoPAC (Figure 1). The AFI feature involves the manual placement of markers on each side of the mitral annulus and left ventricular (LV) apex in three standard apical views. Next, the program automatically tracks the endocardial border and calculates the myocardial (ROI). When necessary, manual adjustments were made to the ROI and/or the endocardial/epicardial borders, which are important for the strain analysis (Figure 1).

For 3D STE analysis, we used the automatic left ventricular quantification function (AutoLVQ) of EchoPAC. Topographic markers were placed in the middle of the mitral valve and the LV apex. The endocardial border was automatically delineated, and manual adjustments were made when necessary. For both 2D and 3D STE, end-diastole and end-systole were determined by automatic identification of the aortic valve opening and closing, and manual adjustments were made when necessary. All the 2D and 3D global longitudinal strain values were calculated using the software, and presented as a 17-segment bull’s eye model (Figure 2).

### 2.3. Statistics

Continuous variables were expressed as mean ± standard deviation, and compared by *t*-test or Mann–Whitney U test, as appropriate. Categorical variables were expressed as percentages, and compared by the chi-square test, Fisher’s exact test, or logistic regression, as appropriate. We investigated the association of 2D GLS with 3D GLS using linear regression analysis, Pearson’s, Spearman correlation, and a Bland-Altman plot. A two-tailed α value of <0.05 was considered statistically significant—except for the test of equality of covariance matrices, for which *p* < 0.005 was considered significant. Normal distribution of the data was assessed by Shapiro–Wilk’s test. The majority of analyses were performed on the entire group of patients, some analyses were performed separately for males and females. All analyses were performed using IBM SPSS for MAC version 23 (IBM, New York, NY, USA) and MS Excel (Redmond, Washington, DC, USA) for MAC version 16.5.

### 2.4. Reproducibility Analysis

Intra- and inter-observer variability of 2D and 3D GLS measurement was tested in all subjects. Intra-observer variability was tested by repeated measurements four or more weeks apart with blinding to the original dataset. To test inter-observer variability, a second experienced operator evaluated the loops with no access to the original dataset. Intra- and interobserver variability is presented as mean percentage error, and it was calculated as an absolute difference between the two measurements.

## 3. Results

Our analysis included a total of 90 patients. Table 1 shows the baseline clinical characteristics, including gender differences. Except for LVEF, heart failure, and CABG, there were no meaningful differences between males and females. Females were less represented in the whole patient sample (41%). Mean value of 2D and 3D GLS was −10.6 ± 4.2 and −10.5 ± 4.1, respectively. A Shapiro–Wilk’s test (*p* > 0.05) and visual inspection of the histograms, normal Q-Q plots and box plots showed that the data (2D, 3D GLS) are approximately normally distributed for both males and females. There were very few outliers in the 2D and 3D GLS dataset. Since they were not due to data entry error, and they do not affect the assumptions made in the analysis or the results, they were not removed from the analysis.

We observed an overall strong positive correlation between 2D GLS and 3D GLS (Pearson’s R = 0.76, *p* < 0.001, Spearman *ρ* = 0.74, *p* < 0.001) (Figure 3). Separate analyses revealed that this correlation coefficient was numerically greater in males (Pearson’s R = 0.78, *p* < 0.001, Spearman *ρ* = 0.75, *p* < 0.001) and lesser in females (R = 0.69, *p* < 0.001, Spearman *ρ* = 0.66, *p* < 0.001), though this difference was not significant. A Bland–Altman analysis demonstrated a small bias (0.1%) and moderate limits of agreement (SD: 2.9%) between 2D and 3D GLS (Figure 4).

Analysis of every 2D vs. 3D segment of the 17-segment AHA model revealed a poor associated segment correlation, with R and *ρ* values ranging from 0.2 to 0.5 (*p* < 0.01) (Table 2). Not all the segments even reached the level of significance. The anteroseptal segments seem to produce a higher correlation between 2D and 3D GLS irrespective whether they were apical, middle, or basal (Table 2). The correlation between 2D and 3D GLS was weaker among individuals with >50% ventricular pacing (R = 0.62, *p* < 0.001) than in individuals with <50% ventricular pacing or no pacing (R = 0.8, *p* < 0.001) (Figure 5*,* Table 3). Moreover, the correlation and regression coefficients between 2D vs. 3D GLS were lower with LVEF < 35% (R = 0.69, *p* = 0.002) than LVEF > 35% (R = 0.72, *p* < 0.001) (Figure 6, Table 3). Other clinical or paraclinical parameters did not influence the level of correlation between 2D and 3D GLS. Intra-observer variability for 2D and 3D GLS was 2 and 2.3%, respectively. Inter-observer variability for 2D and 3D GLS was 3.8 and 3.6%, respectively.

## 4. Discussion

The main findings of our retrospective analysis can be summarized as follows: (1) we found a high agreement between the two-dimensional and the three-dimensional global longitudinal strain, (2) segmental agreement between the 2D and the 3D strain was poor, (3) the degree of agreement differed between genders, though not significantly, and (4) cardiac pacing and reduced LVEF were associated with a lower numerical correlation between the two-dimensional and the three-dimensional global longitudinal strain.

### 4.1. Previous Studies

Two-dimensional speckle-tracking echocardiography has been proven to be efficient and reliable for the quantification of regional and global LV myocardial motion in different clinical scenarios, yet it has some limitations [1,3,4,5,6,9,12]. Three-dimensional speckle-tracking echocardiography has attracted interest because it may overcome the “out of plane” movement limitation of 2D STE. However, the greater complexity of 3D STE acquisition and image analysis make it vulnerable to low image quality, tracking artifacts, and low frame rate interactions. Although it has been shown that 3D STE performance is not compromised by frame rates as low as 18–25 frame/s [14].

Many recent comparative studies have examined 2D STE, 3D STE, and MRI tagging or feature tracking, showing varying results. Altman et al. conducted a trial comparing different 2D STE and 3D STE measures, and they found that GLS was similar between 2D and 3D modes (−14 ± 4 vs. −13 ± 3, non-significant) [15]. Another trial evaluated the agreement between 3D and 2D speckle-tracking GLS, and it found a Pearson correlation of 0.95 [16]. On the other hand, a comparison of 2D vs. 3D GLS detected a correlation coefficient of only 0.4 in healthy volunteers, compared to 0.9 in patients with mitral stenosis [6]. Another study reported a good correlation between GLS determined by cardiac magnetic resonance feature tracking (CMRFT) compared to 2DSTE (r = 0.83) and 3DSTE (r = 0.87) [17]. In one investigation, GLS values were consistently lower in the 3D mode compared to the 2D mode, and the sensitivity for predicting coronary artery disease was 80% for 2D GLS compared to 93% for 3D GLS [18]. Notably, 3D strain data were acquired faster than 2D data (2.2 ± 1 vs. 3 ± 1 min, respectively) [18].

In addition, Mirea et al. [19] showed significant 2D segmental strain variation of up to 4.5%, though the parameters from each vendor correlated to the mean of all vendors ranging from 0.58 to 0.81 [19].

The varying levels of agreement between 2D and 3D strain data may be explained by the different vendors, intra- and interobserver variability, differences in patient comorbidities and gender, cine loop image quality, and the magnitude of manual adjustments.

### 4.2. Current Study

To the best of our knowledge, no prior studies have compared 2D and 3D GLS between segments of the 17-segment AHA model. Surprisingly, although we found a high overall agreement between 2D and 3D GLS, the numerical correlation per segment was quite low. This is probably due to the different manner of acquisition and the segmentation processes. In the 3D mode, the calculation originates from the volumetric matrix. On the contrary, in the 2D mode, an extrapolation is created from three 2D apical planes. Moreover, we may speculate that the strain segment annotation differs between 2D and 3D mode, which could explain why there was a generally strong correlation between the two modes overall, but not per segment. On the other hand, there is some pattern to the level of correlation related to broader areas of the myocardium. Particularly anterior and anteroseptal areas demonstrated higher 2D vs. 3D GLS correlation than the rest of the heart. We may only speculate that better visualization of anterior and septal areas, which are usually in the direct beam of the ultrasound transducer, may lead to more reliable speckle tracking acquisition. On the lateral, posterior, and inferior myocardial wall the tracked border may more easily depart from the visible boundary [20]. A study by Patrianakos et al. showed a good agreement of segmental 2D STE in apical segments and a poor correlation of basal segments obtained using two different echocardiography devices [21].

Low 2D vs. 3D segmental agreement may be also caused by different mechanisms of tracking.

The sensitivity of the 3D tracking is expected to be lower in comparison with 2D tracking due to a lower resolution, thus bigger speckles to track. Also, 3D tracking corresponds to surface shortening as opposed to 2D, which represents linear shortening [22]. Since LV deformation involves a combination of apex-to-base movement, thickening, and simultaneous twisting, speckles exhibit genuine 3D motion, which 2D STE cannot account for as compared to 3D STE [22].

After the publication of the first inter-vendor study demasking a significant difference of 3.7% strain units [4], software adjustments were made to improve the inter-vendor GLS agreement [23]. This could be achieved by giving more weight to the global shortening in detriment to the local displacement.

In our study, we also found that a low LVEF of <35% and a significant amount (>50%) of right ventricular pacing were associated with a decreased correlation between 2D and 3D GLS. Both factors have the same denominator of asynchronous and/or impaired ventricular contraction. It has been also shown that the left ventricle geometry may act as a confounder. A significant reduction of GLS could be compensated by a small increase in global circumferential strain, wall thickness, and/or reduced LV diameter [24]. We may speculate that more complex 3D myocardial motion during pacing and with reduced LVEF may enhance the difference between 2D and 3D tracking based on “out of plane” motion.

Of note, the reproducibility of our GLS measurements was comparable to both of the inter-vendor trials [3,4] but higher than in the trial by Altman et al. [15]. Therefore, it is less likely that intra- or inter-observer variability meaningfully accounted for 2D and 3D strain variability in our trial.

Finally, our study is more hypothesis-generating than completely enlightening the association of GLS and segmental strain data. Comparison of 2D and 3D global and segmental strain data warrants further systematic prospective studies utilizing reference modality (CMR), computer simulated data to begin with and/or different vendor agreement studies.

### 4.3. Implications

Our observations may have implications for clinical practice. The global longitudinal strain is a reliable and a reproducible measure of myocardial deformation, even when assessed using different modes of acquisition (2D vs. 3D). On the contrary, caution should be paid when evaluating segmental strain data, which may significantly differ between 2D and 3D modes of tracking. It seems that GLS is more representative of global shortening than local displacement. In patients with a reduced left ventricular ejection fraction and a significant amount of right ventricular pacing, strain data must be evaluated cautiously.

### 4.4. Limitations

The study has several limitations. First, it was a retrospective study, and many aspects of the acquisition were not prespecified. Moreover, the agreement between the 2D and the 3D strain was determined purely based on echocardiographic methods, without comparison to the “golden standard” reference of MRI tagging or feature tracking. Notably, although there is no real golden standard for myocardial deformation, MRI is historically considered the most accurate and reliable method. In addition, we studied patients with heterogenous cardiovascular diseases.

## 5. Conclusions

We found that 2D and 3D GLS measurements exhibited a close overall agreement, but not when analyzed per segment according to the 17-segment AHA model. It seems that GLS is more representative of global shortening than local displacement. Moreover, high levels of right ventricular pacing and a reduced left ventricular ejection fraction were associated with a numerically lower correlation between 2D and 3D GLS. Therefore, strain data in patients with reduced ejection fraction and right ventricular pacing have to be evaluated with caution.

## Figures and Tables

**Figure 1 jcm-11-02402-f001:**
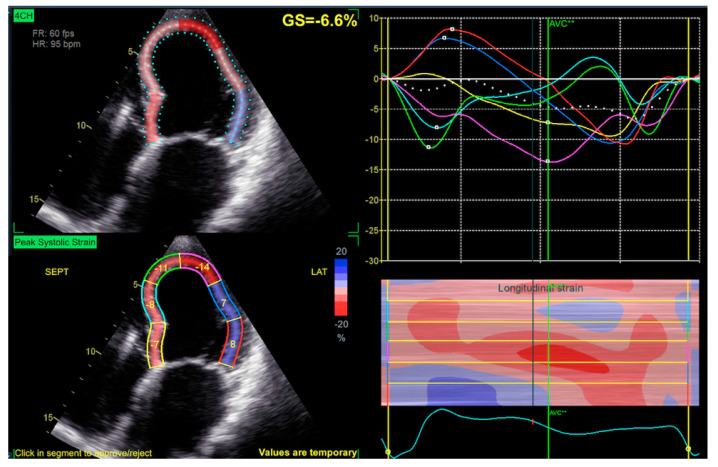
Four–chamber view of automatic function imaging (AFI), peak systolic strain is visualized in the left lower part, global strain in the left upper part and corresponding waveforms in the right upper part, right lower part visualize surface extrapolated color mapped strain.

**Figure 2 jcm-11-02402-f002:**
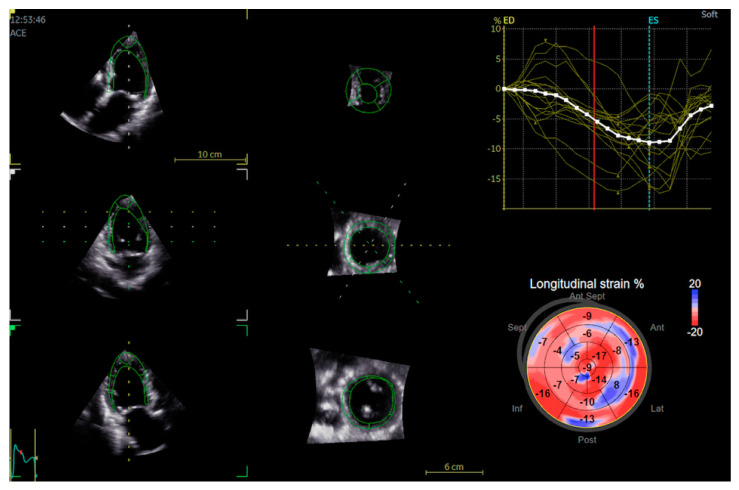
Automatic left ventricular quantification (AutoLVQ) plane after segmentation process with bull’s eye reconstruction of 3D global longitudinal strain. Red color and more negative number means better contractility as opposed to positive numbers and blue colour.

**Figure 3 jcm-11-02402-f003:**
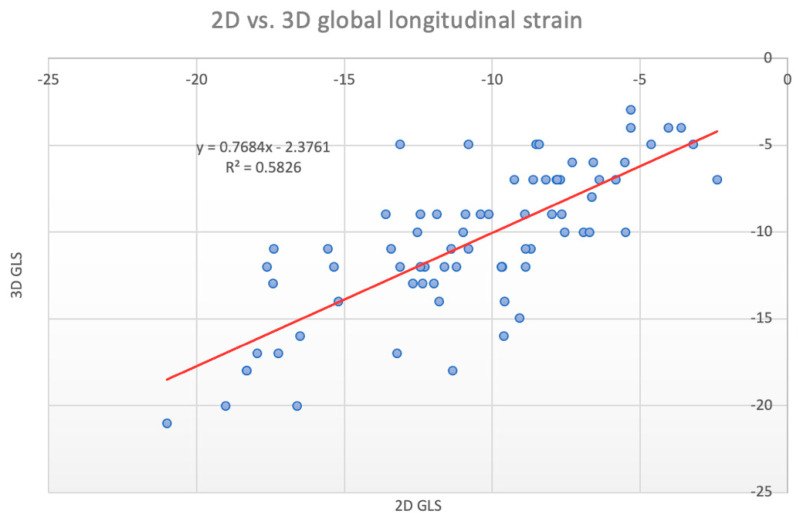
Scatter plot of 2D vs. 3D global longitudinal strain, linear regression equation displayed in the left upper section.

**Figure 4 jcm-11-02402-f004:**
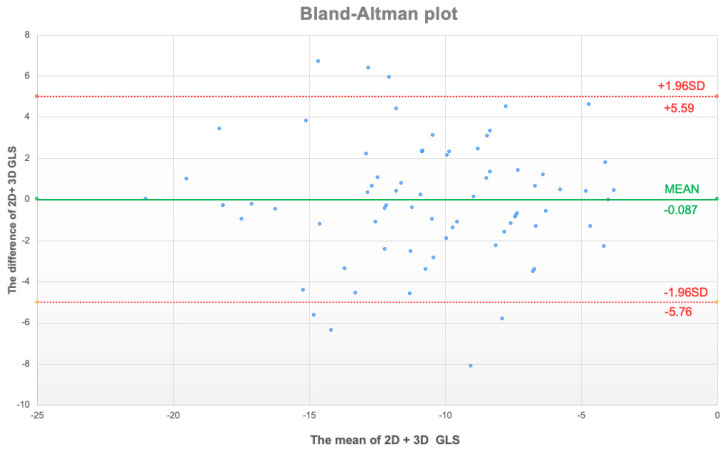
Bland–Altman plot of mean values of 2D + 3D global longitudinal strain (GLS, *x* axis) and the difference between 2D and 3D GLS. Upper and lower limit of agreement displayed as red dotted line with respective values.

**Figure 5 jcm-11-02402-f005:**
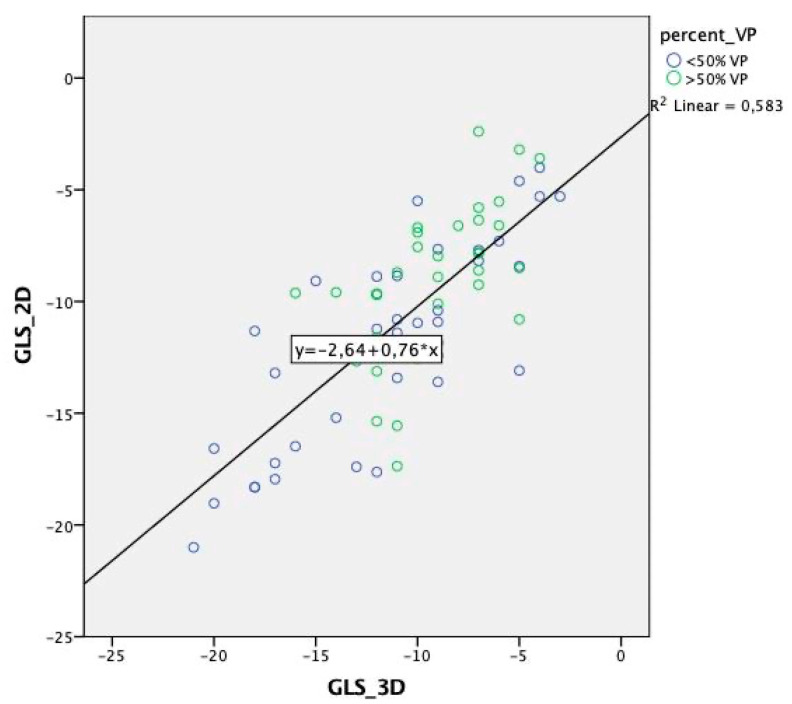
Scatter plot of 2D vs. 3D global longitudinal strain. Sorted by the amount of right ventricular pacing (VP) with displayed regression equation.

**Figure 6 jcm-11-02402-f006:**
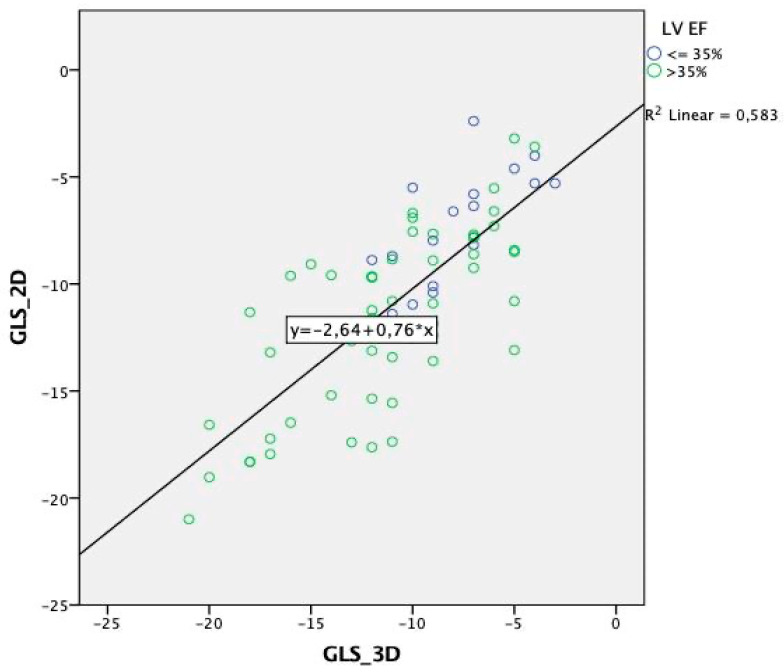
Scatter plot of 2D vs. 3D global longitudinal strain sorted by left ventricular ejection fraction (LV EF) with displayed regression equation.

**Table 1 jcm-11-02402-t001:** Baseline characteristics of the study population.

	Total Population	Males	Females	*p* Value
	N = 90	N = 53	N = 37	
Age (years)	73.2 ± 11.2	70.5 ± 12.3	76.7 ± 8.3	0.681
Males (%)	59	-	-	
Body weight (kg)	85.1 ± 18.4	91.4 ± 16.5	77 ± 17.7	0.882
Body height (cm)	169.1 ± 9.9	175.5 ± 6.4	160.7 ± 7	0.068
Body mass index (kg/m^2^)	29.6 ± 5.4	29.5 ± 4.6	29.8 ± 6.4	0.92
LV EF	48 ± 12.7	45 ± 12.9	51 ± 11.8	0.039
Coronary artery disease (%)	46.5	49	43.2	0.269
Hyperlipoproteinemia (%)	52.6	55.1	48.6	0.726
Myocardial infarction (%)	27.9	32.7	21.6	0.744
Peripheral arterial disease (%)	4.7	8.2	2.7	0.660
Hypertension (%)	79.1	73	86.5	0.371
Heart failure (%)	8.1	18.0	10.8	0.021
Diabetes mellitus (%)	25.6	24.5	27	0.732
Previous stroke/TIA (%)	9.3	10.2	8.1	0.585
COPD (%)	8.1	8.2	8.1	0.314
DCM	8.1	10.2	5.4	0.27
HCM	1.2	2	0	N/A
CABG (%)	16.3	24.5	5.4	0.027
PM (%)	38	32	48	0.656

Indices are shown as mean ± standard deviation or proportion in percentages and compared for male and females.; CABG, Coronary artery by-pass graft; COPD, Chronic obstructive pulmonary disease; DCM, dilated cardiomyopathy; EF, ejection fraction; HCM, hypertrophic cardiomyopathy; LV, left ventricle; N/A, not applicable; PM, pacemaker; TIA, transient ischemic attack.

**Table 2 jcm-11-02402-t002:** Comparison of 2D vs. 3D global longitudinal strain, analyzed on entire group and separately by gender.

	LINEAR REGRESSION + PEARSON’S AND SPEARMAN CORRELATION																				
2D vs. 3D GLS	ALL							MALES							FEMALES						
	SLOPE ± SEM	R	*p*	r	*p*	ρ	*p*	SLOPE ± SEM	R	*p*	r	*p*	ρ	*p*	SLOPE ± SEM	R	*p*	ρ	*p*	r	*p*
	0.77 ± 0.27	0.76	<0.001	0.76	<0.001	0.74	<0.001	0.82 ± 0.35	0.78	<0.001	0.78	<0.001	0.75	<0.001	0.69 ± 0.39	0.69	<0.001	0.66	<0.001	0.69	<0.001
Seg.1	0.38 ± 0.39	0.27	0.019	0.27	0.02	0.32	0.005	0.36 ± 0.36	0.16	0.3	0.18	0.23	0.42	0.006	0.26 ± 0.57	0.15	0.41	0.23	0.218	0.16	0.40
Seg.2	0.66 ± 0.52	0.55	<0.001	0.55	<0.001	0.50	<0.001	0.79 ± 0.51	0.54	<0.001	0.54	<0.001	0.57	<0.001	0.58 ± 1.03	0.58	<0.001	0.47	0.009	0.58	<0.001
Seg.3	0.12 ± 0.59	0.18	0.123	0.20	0.09	0.23	0.053	0.29 ± 0.74	0.3	0.05	0.32	0.04	0.36	0.018	0.09 ± 0.91	0.07	0.715	0.06	0.740	0.08	0.682
Seg.4	0.46 ± 0.60	0.35	0.002	0.35	0.002	0.35	0.002	0.46 ± 0.82	0.35	0.023	0.35	0.02	0.34	0.027	0.44 ± 089	0.33	0.075	0.30	0.104	0.32	0.086
Seg.5	0.42 ± 0.68	0.318	0.006	0.30	0.009	0.35	0.002	0.38 ± 0.79	0.26	0.096	0.24	0.119	0.25	0.116	0.50 ± 1.19	0.40	0.03	0.46	0.011	0.39	0.03
Seg.6	0.05 ± 0.60	0.068	0.569	0.04	0.733	0.12	0.327	−0.01 ± 0.65	0.05	0.750	0.01	0.970	0.13	0.413	0.06 ± 1.11	0.06	0.745	0.06	0.754	0.06	0.753
Seg.7	0.50 ± 0.43	0.44	<0.001	0.44	<0.001	0.48	<0.001	0.43 ± 0.60	0.42	0.006	0.42	0.004	0.51	0.001	0.61 ± 0.57	0.432	0.019	0.49	0.007	0.43	0.016
Seg.8	0.44 ± 0.48	0.47	<0.001	0.47	<0.001	0.56	<0.001	0.59 ± 0.54	0.53	<0.001	0.53	<0.001	0.56	<0.001	0.33 ± 0.88	0.42	0.024	0.51	0.004	0.41	0.02
Seg.9	0.50 ± 0.47	0.41	<0.001	0.41	<0.001	0.33	0.005	0.57 ± 0.59	0.49	0.001	0.49	0.001	0.45	0.003	0.40 ± 0.76	0.32	0.086	0.22	0.251	0.31	0.09
Seg.10	0.32 ± 0.52	0.27	0.02	0.27	0.02	0.35	0.002	0.59 ± 0.65	0.53	<0.001	0.52	<0.001	0.62	<0.001	−0.1 ± 0.85	0.04	0.850	0.631	0.091	0.05	0.794
Seg.11	0.531 ± 0.528	0.45	<0.001	0.45	<0.001	0.50	<0.001	0.63 ± 0.63	0.51	<0.001	0.50	<0.001	0.55	<0.001	0.36 ± 0.91	0.36	0.05	0.31	0.1	0.37	0.04
Seg.12	0.50 ± 0.52	0.45	<0.001	0.45	<0.001	0.39	0.001	0.60 ± 0.59	0.54	<0.001	0.54	<0.001	0.51	0.001	0.39 ± 0.92	0.36	0.05	0.28	0.229	0.37	0.04
Seg.13	0.29 ± 0.65	0.48	<0.001	0.48	<0.001	0.58	<0.001	0.37 ± 0.75	0.51	<0.001	0.51	<0.001	0.62	<0.001	0.16 ± 1.15	0.38	0.05	0.49	0.007	0.37	0.04
Seg.14	0.46 ± 0.66	0.52	<0.001	0.52	<0.001	0.50	<0.001	0.56 ± 0.75	0.56	<0.001	0.57	<0.001	0.56	<0.001	0.39 ± 1.18	0.49	0.007	0.43	0.017	0.49	0.006
Seg.15	0.23 ± 0.61	0.32	0.007	0.26	0.03	0.31	0.007	0.28 ± 0.75	0.42	0.006	0.31	0.05	0.37	0.016	0.17 ± 1.01	0.23	0.220	0.19	0.376	0.2	0.291
Seg.16	0.14 ± 0.56	0.19	0.103	0.18	0.129	0.135	0.257	0.33 ± 0.75	0.45	0.003	0.42	0.006	0.36	0.018	−0.15 ± 0.76	0.17	0.359	0.25	0.190	0.17	0.361
Seg.17	0.27 ± 0.49	0.3	0.01	0.29	0.01	0.26	0.028	0.42 ± 0.66	0.47	0.002	0.44	0.003	0.40	0.773	0.03 ± 0.71	0.04	0.830	0.05	0.773	0.03	0.877

Indices are shown as mean ± standard error of the mean (SEM);2D, Two-dimensional;3D, Three-dimensional; GLS, Global longitudinal strain; values of Pearson’s correlation coefficient (r) and linear regression coefficient (R) coincide while the data are in the same units, thus “naturally” normalized; ρ stands for Spearman correlation.

**Table 3 jcm-11-02402-t003:** 2D and 3D global longitudinal strain, subgroup analysis.

	N	R	*p* Value
LV EF < 35%	17	0.699	0.002
LV EF > 35%	73	0.727	<0.001
VP > 50%	41	0.62	<0.001
VP < 50%	49	0.8	<0.001

EF, Ejection fraction; LV, Left ventricle; VP, Ventricular pacing, R—person’s correlation coefficient.

## Data Availability

The data that support the findings of this study are available from the corresponding author upon reasonable request and with compliance to the General Data Protection Regulation.

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
