# Peer review of "The Agreement of a Two- and a Three-Dimensional Speckle-Tracking Global Longitudinal Strain"

_jcm, 2022, doi:10.3390/jcm11092402_

Round 1

Reviewer 1 Report

In their manuscript the authors present their findings of a retrospective study of 90 patients with normal, reduced LVEF or ventricular pacing.  They performed both 2D STE and 3DSTE with vendor specific software.  The authors aimed to analyze the level of agreement between GLS measured by 2D vs. 3D STE, gender differences in 2D vs. 3D GLS analysis, and factors influencing the level of agreement between 2D vs. 3D GLS.

The authors conclude 2D and 3D GLS measurements exhibited a close association overall, but not when analyzed per segment. Ventricular pacing and low LVEF were associated with a numerically lower correlation.

The study is interesting, however the results are not novel and I am not sure of their clinical relevance.  Previous studies have shown discrepancies between 2DSTE and 3DSTE.

  1. Perhaps the authors could  explore the prognostic role of 3DTSE over 2DTSE.  
  2. Even for 2DSTE segmental reproducibility is not optimal, therefore it makes sense this could apply for 3DSTE.
  3. 3DSTE segmental analysis could be of interest in relation to areas of fibrosis. do any of the patients have cardiac magnetic resonance? Is there a difference in strain values of 2D vs 3D in areas of fibrosis.

Author Response

We thank the reviewer for his valuable suggestions.

We tried to revise the manuscript accordingly, though in many aspects we do not have the data to apply your suggestions.  Even though we believe that the article is now significantly improved. The changes are highlighted by the red font in the manuscript.  Please find below the corresponding point-by-point response.

Comment 1:  Perhaps the authors could  explore the prognostic role of 3DTSE over 2DTSE.  

Response 1: That is a great suggestion and clinically relevant, though none of our patients was systematically followed for cardiovascular events, only re-do echocardiography for reproducibility analysis. In addition the aim of the trial was to compare correlation and reproducibility of the 2DSTE and 3DSTE methods. We are truly sorry, but we cannot ad this information to the results based on our data limitations.

Comment 2: Even for 2DSTE segmental reproducibility is not optimal, therefore it makes sense this could apply for 3DSTE.

Response 2: Completely agree, we added new paragraph (and new references) in the discussion section on variability and reproducibility of the segmental strain measurement and comparison of segmental data between vendors.

Comment 3:   3DSTE segmental analysis could be of interest in relation to areas of fibrosis. do any of the patients have cardiac magnetic resonance? Is there a difference in strain values of 2D vs 3D in areas of fibrosis.

Response 3: We do agree, that correlation of cardiac magnetic resonance regional data and 3DSTE segmental analysis would be of interest. However we dont have magnetic resonance imaging data from our patients but we highligted this as a disadvantage in the limitations section.

Comment 4: The study is interesting, however the results are not novel and I am not sure of their clinical relevance.  Previous studies have shown discrepancies between 2DSTE and 3DSTE.

Response 4: We thank you for your appreciation of our results. We  tried to add some clinicaly relevant outputs to the conclusion section as you have suggested. You are completely right that comparison of 2DSTE and 3D STE are not novel (we refer to the previous studies ). However 2D and 3D segmental data were not compared so far.

Reviewer 2 Report

Dear Authors,

The idea of the study is interesting and  but this study have important  drawback which,  I consider  important  to be corrected

I have added a few specific comments:

1) The title sould be redone

2) The aim of the study sould be more cleare written. The lines 78,79 could be written to the discution section.

3) There aren't the inclusion and exclusion criteria

4) The studied group is various regarding their pathology. This aspect should be mentioned to the limitations.

5) It is not mentioned whether the studied group has a gaussian distribution or a non-gaussian distribution. (Normality tests are not mentioned, for example Kolmogorov-Smirnov test or Shapiro-Wilk test)

6) Regarding the Pearson correlation coefficient, its use is not explained. In situations including not evenly distributed variables, Spearman coefficient could be more reliable. For a higher statistic power, both Pearson and Spearman coefficients could be used.

7) Regarding the linear regression test, we can observe outliers that can alter the results. It is not mentioned whether these variables were excluded or not.

9) The conclusion section should be improved.

In conclusion, I recommend accepting the article for publication at the IJERPH journal with major revision.

Author Response

We thank the reviewer for his valuable suggestions.

We revised the manuscript thoroughly and fully according to your recommendations. We believe that the article is now significantly improved. The changes are highlighted by the red font in the manuscript.  Please find below the corresponding point-by-point response.

Comment 1:  The title sould be redone

Response 1: We tried to be specific about the results already in the title of the manuscript, but i assume it may feel too „wordy“. We radically shortened the title of the article.

Comment 2: The aim of the study should be more cleare written. The lines 78,79 could be written to the discution section.

Response 2: The aim of the study was re-written, we do believe it is more clear now. The lines 78,79 related to gender-related influences on left ventricular mechanics are in our opinion justifying the need for separate analysis for males and females. That is the sole reason why we prefer it in the introduction section though it may be felt as a subject to discussion section. If you agree, we will leave it in the introduction section, otherwise we may also place it as you rcommended in the discussion section.

Comment 3:  There aren't the inclusion and exclusion criteria

Response 3: Since the study was retrospective in nature and consecutive patients were included, we have not mentioned inclusion/exclusion criteria separately. The only exlusion criteria were incomplete loops, inadequate image quality and atrial fibrillation at the time of image aquisition as mentioned in materials and methods. We have re-written this section to make this statement more obvious.

Comment 4: The studied group is various regarding their pathology. This aspect should be mentioned to the limitations.

Response 4: We do agree, important point. We added this aspect to the limitations section.

Comment 5:  It is not mentioned whether the studied group has a gaussian distribution or a non-gaussian distribution. (Normality tests are not mentioned, for example Kolmogorov-Smirnov test or Shapiro-Wilk test)

Response 5: Good point, thank you. A sentence was added to the methods section and a paragraph to the results section explaining the normal distribution of the data (2D,3D GLS).

Comment 6: Regarding the Pearson correlation coefficient, its use is not explained. In situations including not evenly distributed variables, Spearman coefficient could be more reliable. For a higher statistic power, both Pearson and Spearman coefficients could be used.

Response 6: We believe Pearson correlation was apropriate given the character of the GLS data (interval variables, distribution).However,  Spearman correlation may bring more statistical power as you suggested,  thus we added new calculations into the results section and have rewrittent a refitted table  to accomodate the results of Spearman rho and its respective P values.

Comment 7: Regarding the linear regression test, we can observe outliers that can alter the results. It is not mentioned whether these variables were excluded or not.

Response 7: First, there were very few true outliers. (as we see it per definition, since the term may be perceived quite widely among both scholars and clinicians) outliers defined as a value > upper boundary and < lower boundary; upper boundary defined as 3rd quartile+(1.5*interquartile range) and lower boundary defined as 1st quartile –(1.5*interquartile range). The outliers in our study were not due to data entry error (we inspected this thoroughly), they also do not affect the assumptions made in the analysis nor the results of the analysis. The outliers were thus not removed from the analysis. For more clarity we added a paragraph in the results section stating the above mentioned as you have recommended.

Comment 8:  The conclusion section should be improved.

Response 8: We tried our best to improve also this section and bring some clinical conclusion.

Round 2

Reviewer 2 Report

I consider that the authors have improved sufficiently the manuscript. In this form, after minor revision the paper can be accepted.